# Exploring the Etiological Links behind Neurodegenerative Diseases: Inflammatory Cytokines and Bioactive Kynurenines

**DOI:** 10.3390/ijms21072431

**Published:** 2020-03-31

**Authors:** Masaru Tanaka, József Toldi, László Vécsei

**Affiliations:** 1MTA-SZTE, Neuroscience Research Group, Semmelweis u. 6, H-6725 Szeged, Hungary; tanaka.masaru.1@med.u-szeged.hu; 2Department of Neurology, Interdisciplinary Excellence Centre, Faculty of Medicine, University of Szeged, Semmelweis u. 6, H-6725 Szeged, Hungary; 3Department of Physiology, Anatomy and Neuroscience, University of Szeged, Közép fasor 52, H-6726 Szeged, Hungary; toldi@bio.u-szeged.hu

**Keywords:** neurodegenerative diseases, Alzheimer’s disease, Parkinson’s disease, neuroinflammation, amyloid, tau, prion, cytokine, tryptophan, kynurenine, multifactorial causality

## Abstract

Alzheimer’s disease (AD) and Parkinson’s disease (PD) are the most common neurodegenerative diseases (NDs), presenting a broad range of symptoms from motor dysfunctions to psychobehavioral manifestations. A common clinical course is the proteinopathy-induced neural dysfunction leading to anatomically corresponding neuropathies. However, current diagnostic criteria based on pathology and symptomatology are of little value for the sake of disease prevention and drug development. Overviewing the pathomechanism of NDs, this review incorporates systematic reviews on inflammatory cytokines and tryptophan metabolites kynurenines (KYNs) of human samples, to present an inferential method to explore potential links behind NDs. The results revealed increases of pro-inflammatory cytokines and neurotoxic KYNs in NDs, increases of anti-inflammatory cytokines in AD, PD, Huntington’s disease (HD), Creutzfeldt–Jakob disease, and human immunodeficiency virus (HIV)-associated neurocognitive disorders, and decreases of neuromodulatory KYNs in AD, PD, and HD. The results reinforced a strong link between inflammation and neurotoxic KYNs, confirmed activation of adaptive immune response, and suggested a possible role in the decrease of neuromodulatory KYNs, all of which may contribute to the development of chronic low grade inflammation. Commonalities of multifactorial NDs were discussed to present a current limit of diagnostic criteria, a need for preclinical biomarkers, and an approach to search the initiation factors of NDs.

## 1. Introduction

Neurodegenerative diseases (NDs) encompass a range of progressive and often incurable conditions which affect the central nervous system (CNS), leading to neurodegeneration and eventually to neural cell death, which cause a broad spectrum of symptoms from motor dysfunctions to psychobehavioral manifestations such as ataxia and dementia [1]. Alzheimer’s disease (AD) and Parkinson disease’s (PD) are the most common NDs. AD represents approximately 60%–70% of about 50 million people worldwide suffering from dementia, which is a major cause of disability and dependency among the elderly. More than 10 million people worldwide suffer from PD, and the incidence increases with age [2]. Other NDs include multiple sclerosis (MS), Huntington’s disease (HD), amyotrophic lateral sclerosis (ALS), Creutzfeldt–Jakob disease (CJD), human immunodeficiency virus (HIV)-associated neurocognitive disorders (HAND), and stroke-induced secondary neurodegeneration (SND), among others. [3]. Neurological diseases are the leading causes of disability-adjusted life years and second leading cause of deaths after cardiovascular diseases globally [2]. The names of the diseases labeled in honor of distinguished physicians, subtyping according to pathological and clinical findings, overlapping histopathological classification, and numerous molecular discoveries all obscure a fundamental perspective of the diseases and may hamper finely tuned research into pathogenesis and new drug discovery of NDs [4,5,6,7,8]. 

Neurodegeneration, a structural and/or functional loss of neurons in the CNS, is a common pathognomonic finding of the brain regions reflecting impairment and dysfunction of the corresponding motor, autonomic, and/or cognitive nervous systems [9]. The neurodegenerative lesions of postmortem brain specimens correlate well with structural and functional imaging studies. The regional patterns of the brain shrinkage may help identify affected domains and diagnose diseases by magnetic resonance imaging (MRI) and positron emission tomography (PET) [10].

Abnormal accumulation of proteinaceous materials in and around neurons is a histopathological hallmark in the degenerative neural tissues. Amyloid-β (Aβ) protein, tau protein, alpha-synuclein (α-syn), transactive response DNA-binding protein (TDP)-43, and myelin debris are common deposits [7]. Most NDs present the inflammatory reactions by activation of the innate immune response factors, as well as the adaptive immune response factors. Neuroinflammation is at least one of the final common pathways leading to neurodegeneration. In degenerative brain tissues, activated glial cells and astrocytes cause accumulation of the abnormal proteins, potentiating further neural injuries [11].

This review article overviews the molecular, pathological, and clinical findings of NDs including neurodegeneration, which is caused by histopathological insults and responsible for specific signs and symptoms, and neuroinflammation which is potentiated by neurodegeneration and exacerbating proteinopathies. A literature search was employed in PubMed/MEDLINE, using appropriate search terms and filters according to a theme of each section. The systematic reviews were conducted to synthesize human sample studies on pro-inflammatory and anti-inflammatory cytokines to present the status of inflammation, and on bioactive kynurenines (KYNs) to examine the involvement of KYN metabolites in major NDs, as described in detail in Appendix A. Finally, commonalities of the NDs were discussed to present a current limit of diagnostic histopathological techniques, a need for preclinical and prodromal diagnostic biomarkers to prevent disease progression, and a possible approach toward discovery of new biomarkers and drugs.

## 2. The Common Theme: Proteinaceous Deposits, Neurodegeneration, Neuropathies 

Neurodegeneration in the CNS is responsible for signs and symptoms of healthy aging and NDs [9]. The senescent and pathological processes can be observed in multiple levels of neurotransmission ranging from molecular to cellular to systemic [12]. Abnormal deposits of proteinaceous materials, their cellular and anatomical distribution, and their subsequent neurodegeneration constitute the pathological criteria essential for neuropathologic diagnosis [7]. 

AD is the most common chronic ND with an insidious onset of progressive cognitive dysfunctions, especially memory impairment. In the early stage, there are no motor or sensory dysfunctions. Motor and autonomic dysfunctions are associated with the comorbidities and typical to the later stages [13]. Anxiety is prevalent, in addition to apathy, depression, aggression, and sleep disorder [14]. Atrophy of frontal, temporal, and parietal cortices, enlargement of the temporal horn of the lateral ventricle, and atrophy of the entorhinal cortex, amygdala, and hippocampus are characteristic findings in patients with AD [15]. AD is the most frequent type of amyloidosis in which the amyloids, insoluble fibrous proteins, abnormally deposit in the neurons and glia. The morphology and neuroanatomic locations of the amyloid deposit are characteristic to AD and stage of the disease [7]. A deposit of Aβ is mainly located in the neocortex and hippocampus of patients with AD. The neuroanatomical distribution of Aβ deposit correlates well with the cognitive symptoms and progression to psychobehavioral manifestations [16]. 

The Aβ deposition leads to microglial activation, cytokine release, reactive astrocytosis, and an induction of inflammation [17]. Aβ oligomers also cause the proteasome-dependent degradation of cadherin 1 (Cdh1), which downregulates glutaminase [18]. Increased glutamate causes a sustained low-level activation at the glutamate receptors, including *N*-methyl-d-aspartate (NMDA) receptors. The chronic excitotoxic insult leads to neuronal death and cognitive impairment [19]. Aβ oligomers directly trigger Ca^2+^ flux through the plasma membrane to increase intracellular Ca^2+^ concentration, leading to mitochondrial Ca^2+^ overload, superoxide radicals-induced oxidative stress (OS), and pro-apoptotic mitochondrial proteins production [20] 

Tauopathy is highly prevalent in AD, characterized by abnormal accumulation of tau protein in the neurons and glia. It is reported that tau deposits better correlate with the severity of cognitive symptoms [7,21]. However, the tauopathy of AD is considered secondary to the deposition of Aβ [22]. OS leads to abnormally increased phosphorylated tau proteins polymerizing to form neurofibrillary tangles. Tau proteins play an important role in the assembly of microtubules and stability of microtubules network in neurons. The dysfunction of tau proteins affects the structural and regulatory functions of the cytoskeleton, leading to abnormal axonal transport, synaptic dysfunction, impaired neuroplasticity, and neurodegeneration [23].

PD is a progressive nervous system disorder that affects movement with symptoms including muscle rigidity, tremors, and changes in speech and gait. In the early stages of PD, main motor dysfunctions include bradykinesia, resting tremors, and rigidity, which appear and largely depend on dopaminergic nigrostriatal denervation [24]. Patients with PD may experience several psychobehavioral symptoms including apathy, agitation, hypersexuality, and pathological gambling, among others. Psychoses, hallucinations, depression, and anxiety are not rare, and symptoms can be present in early stages of the disease, sometimes even before the appearance of classical motor symptoms [25]. Generalized anxiety disorder is common but often unrecognized [26]. Loss of neurons and gliosis of the pars compacta of the substantia nigra and the presence of Lewy bodies (LBs), eosinophilic to basophilic concentric structures with peripheral halos, in pigment nuclei such as the nucleus basilaris, are hallmarks of PD. LBs contain the abnormal aggregates of misfolded α-syn which was discovered as a non-amyloid component of the senile plaques in AD. Mutations in the gene for α-syn were then discovered in familial PD. Pathological aggregations of α-syn are called synucleinopathies [7]. Accumulation of aggregated α-syn in oligodendrocytes forms glial cytoplasmic inclusions, characteristic to multiple system atrophy, which lacks LB pathology. The mechanisms that govern α-syn fibrillization and LB formation in the brain remain poorly understood [27]. 

MS is an autoimmune demyelinating disease in which nerve cells in the brain and spinal cord are damaged. More common symptoms of MS range widely from motor and autonomic dysfunctions to psychobehavioral disturbances, such as gait difficulties, paresthesia, spasticity, vision problems, dizziness and vertigo, incontinence, constipation, sexual problems, pain, cognitive and emotional changes, and depression. Meta-analysis displayed a high prevalence of anxiety and depression in MS patients [28]. The neural lesion of MS is represented by numerous glial scars, called plaques which develop in the white matter and spinal cord [29]. Active plaques are characterized by activated mononuclear cells such as lymphocytes, microglia, and macrophages which destroy myelin and oligodendrocytes into myelin debris, proteins, and lipids. Consequently, the plaques develop demyelinated axons traversing glial scar tissue, forming inactive plaques. Partially remyelinated plaques by oligodendrocytes are shadow plaques. The plaques inactive in their center but expanding at their periphery are smoldering plaques [30].

HD is a fatal autosomal-dominant disease that causes progressive and irreversible motor dysfunctions, resulting in coordination and gait difficulties, as well as cognitive and behavioral changes. Autonomic symptoms such as orthostatic hypotension, excessive perspiration, and tachycardia can occur in mild HD, and the vegetative symptoms are most prominent in the advanced stages [31]. Anxiety is common to HD with the prevalence ranging from 13% to 71%, with no difference between symptomatic patients and pre-symptomatic HD gene carriers [32]. Degeneration and neural loss of the striatum, particularly the caudate nuclei, targeting the cerebral cortex, pallidum, thalamus, brainstem, and cerebellum, are specific neuropathological findings in HD [33]. The degree of the pathological changes correlates with that of disability. An abundance of ballooned neurons in the cerebellum, thalamus, and brain stem were observed [7]. Mutant huntingtin protein is associated with ballooning cell death (BCD), in which cells swell like a balloon until their membrane ruptures. BCD and resultant necrosis are triggered in a mechanism of transcriptional repression-induced atypical cell death of neuron (TRIAD) with reduced levels of a transcriptional co-activator yes-associated protein (YAP) and transcriptional enhancer factor (TEF) [34].

ALS is a degenerative disease leading to the dysfunction of neurons controlling voluntary muscles. ALS often begins with fasciculation, myasthenia, or dysarthria. Later, it involves the muscles responsible to move, speak, eat, and breathe [35]. ALS patients present mild symptoms affecting the autonomic nervous system including urinary urgency, constipation, and sudomotor, as well as the gastrointestinal and cardiovascular systems, and neuropsychiatry [36,37]. Anxiety disorders and depression were reported in ALS patients [38]. TDP-43 proteinopathy results from abnormal inclusion of a nuclear protein in the cytoplasm, nucleus, and cell processes. TDP-43 is a major component of protein aggregates in ALS, found in the lower motor neurons in the spinal cord and brainstem and the upper motor neurons in the motor cortex. TDP-43 can also be found in the hippocampus, amygdala, and cortex in the late stage ALS and ALS patients with dementia [7]. TDP-43 is an RNA/DNA-binding protein with considerably high sequence specificity. TDP-43 proteinopathy is characterized by mislocalization of the nucleus to the cytoplasm, including body formation of ubiquitinated and hyper-phosphorylated TDP-43, truncated toxic C-terminal TDP-43 fragment formation, and protein aggregation [39]. 

The initial symptoms of CJD are rapidly progressive dementia including amnesia, personality changes, and hallucinations with myoclonus. Psychobehavioral symptoms include depression, anxiety, paranoia, and psychosis, whereas motor symptoms include speech impairment, ataxia, and rigid posture [40]. Cortical atrophy such as neural loss and gliosis may be observed, associated with the duration of the disease. The spongiform change in the gray matter with fine and coalescent vacuoles in the neurons and astrocytosis are characteristics of CJD [7]. Generated by proteolysis, fragments of soluble cellular prion protein (PrP^C^) function in maintaining myelin homeostasis and inducing neurite outgrowth [41]. The pathomechanism of prion diseases is the scrapie isoform of the prion protein (PrP^Sc^) template-directed misfolding of PrP^C^ into a pathogenic, conformationally altered, β sheet–rich version of its own protein. PrP^Sc^ accumulation disturbs the ubiquitin/proteasome system, affects the autophagy/lysosome pathway, and exhausts the unfolded protein response pathways in early disease states, weakening the neurons, causing synaptic loss, and inducing cell death pathways [42].

HAND refers to CNS complications from late HIV-1 infection, characterized by the deterioration of the immune system, leading to opportunistic infections and malignancies. Progressive dementia accompanied by motor and behavioral impairments is a representative symptom. Main motor symptoms are ataxia and tremors. In the latent phase, autonomic dysfunction including orthostatic syncope, hypotension, and diarrhea may appear in an autoimmune reaction to HIV. Cognitive impairment often appears as memory loss, dysarthria, concentration difficulty, and poor judgment, and anxiety and mood swings are common [43]. The postmortem autopsies and imaging studies showed striking cortical atrophy like that of AD. Multinucleated and mononuclear macrophages are more prominent in white than gray matter and budding virions were shown from multinucleated cells. Brain infarcts and areas of ischemia, hemorrhage, and large-vessel atherosclerosis are common findings in the HIV-infected elderly on combination antiretroviral therapies (cART) [44].

SND is caused secondary to insufficient blood flow in the brain, presenting symptoms beyond the original insults such as paresthesia, arthralgia, ataxia, blindness, and unconsciousness. Autonomic nervous system (ANS) symptoms are arrythmia, blood pressure irregularity, asymmetric sweating, bladder and bowel dysfunction, and impotence [45]. Anxiety is more frequent than depression in the acute stage of ischemic stroke [46]. There is a strong link among depression, anxiety, and stroke. Clear AD-like pathological changes were observed in SND and combined vascular disease and AD.

Abnormal protein deposits observed as in amyloidosis, tauopathy, synucleinopathy, and/or TDP-43 proteinopathy are inextricably linked to the pathogenesis and progression of neurodegeneration, which presents anatomically corresponding motor, sensory, and/or autonomic dysfunctions. The molecular alterations in concert with activation of microglia and astrocytes induce the release of pro-inflammatory cytokines, leading to an increased production of highly reactive oxygen and nitrogen species (ROS and RNS), which further damage the surrounding tissues and activate the surrounding glial cells. Cognitive domains are largely affected, and psychobehavioral symptoms such as depression and anxiety are common findings in NDs (Figure 1a,b).

## 3. Neuroinflammation: A Common Prelude to Neurodegeneration

Chronic CNS inflammation mediated by microglial cells and astrocytes is a common feature among NDs. The pro-inflammatory cytokine-activated microglial cells, peripherally derived monocytes, and macrophages recruited through the blood–brain barrier (BBB) are main players in inflammation. Microglial cells are the main source of the pro-inflammatory cytokines in the brain [47]. Microglial cells directly respond to the corticosterone peak by expressing both glucocorticoid (GC) and mineralocorticoid receptors. Corticosterone affects the microglial cells though the hippocampus and prefrontal cortex abundant in the GC receptors [48]. 

Activated microglial cells characterized by thick, shorter processes, and swollen cell bodies produce more cytokines, recruiting the peripheral monocytes. Both microglial cells and peripheral monocytes secrete pro-inflammatory cytokines such as interleukin (IL)-1β, tumor necrosis factor (TNF)-α, and IL-6 [49].

Predisposing factors of neuroinflammation are aging, metabolic diseases, hypertension, stroke, depression, and dementia, among others. Healthy aging is associated with chronic inflammation, which contributes to increased vulnerability to anxiety and depression. Adipose tissue dysfunction in obesity and hypertension contribute to chronic low-grade inflammation which predisposes to the development of insulin resistance and cardiovascular disease [50]. Cerebral small-vessel disease and atherosclerosis induce microvascular hypoperfusion, oligodendrocyte malfunction, and demyelination, leading to perpetual low-grade inflammation, which amplifies the risk of stroke. Viral infection causes neuroinflammation by hematogenous dissemination through BBB invasion and retrograde dissemination through nerve cells [51]. HIV infection is associated with increased cytokine levels, cholesterol, lipopolysaccharides, apolipoprotein E4, insulin resistance, and testosterone deficiency, all of which contribute to neuroinflammation. HIV-1 can replicate in macrophages and R5 T lymphocytes, both of which are independently associated with dementia [52]. Aging is a prevalent risk factor of metabolic diseases, hypertension, depression, and dementia [53] (Figure 1c,e).

### 3.1. Systematic Reviews on Inflammatory Cytokines in Neurodegenerative Diseases

A systematic review was conducted to study the status of pro- and anti-inflammatory cytokines in NDs. Selection criteria and risk of bias assessment are described in Appendix A. A total of 23,206 articles were included in our database search. Then, 252 meta-analyses and systematic reviews and 342 articles were evaluated for eligibility. Finally, 22 articles were eligible for the systematic review (Figure A1). The articles included for the synthesis, study types, and risk of bias assessment are presented in Table A1. Evidence levels were evaluated at low risk of bias for AD, PD, ALS, and CJD, but high risk of bias for MS, HD, HAND, and SND (Table A1).

All NDs present altered inflammatory reactions evidenced by activation of the innate immune response factors and pro-inflammatory cytokines, as well as the adaptive immune response factors and anti-inflammatory cytokines. The alterations of cytokine levels were observed in AD patients both with the plasma and cerebrospinal fluid (CSF). Pro-inflammatory IL-1β, IL-6, and TNF-α, as well as anti-inflammatory cytokines, IL-1 receptor antagonist, and IL-10, are elevated both in the CSF and in the plasma of AD patients [54]. C–C motif chemokine ligand 5 (CCL5) is elevated in AD. C–X–C motif chemokine 12 is negatively associated with AD. The serum levels of IL-2 are elevated in the early stage of AD [55]

Both innate and adaptive immune activations are observed in PD. Meta-analysis of cytokine status in blood samples of PD patients showed higher levels of pro-inflammatory cytokines IL-1β, IL-6, TNF-α, c-reactive protein, and CCL5, and anti-inflammatory cytokines IL-2 and IL-10 in the serum of PD patients. No significant difference was observed in interferon (IFN)-γ, IL-4, and IL-8 levels [56]. Meta-analysis of CSF samples of PD patients reported significantly higher levels of IL-1β, IL-6, and transforming growth factor (TGF)-β1 [57]. Pro-inflammatory cytokines, IL-1, IL-12, IL-17, IL-22, TNF-α, and interferon (IFN)-γ are elevated in MS, possibly contributing to the demyelinating of the neural pathways. In contrast, anti-inflammatory cytokines such as IL-4 and IL-10 are lower in MS [58].

The early stage of HD is characterized by microglial activation and neuroinflammation [59]. Of microglia-derived inflammatory markers IL-6, matrix metallopeptidase 9, vascular endothelial growth factor (VEGF), and TGF-β1 levels were significantly increased, while IL-18 level was significantly reduced in plasma of HD. Plasma IL-6 was reversely correlated with the severity of HD [60]. A two-year follow-up study showed increased TNF-α and IL-10 levels. Higher IL-6 and lower IL-5 levels were observed in HD patients with motor symptoms than premotor ones. IL-6 and IL-8 levels were inversely associated with HD progression [61]. A post-mortem study revealed increased expression of inflammatory mediators, including C–C motif chemokine ligand 2 and IL-10 in the striatum of HD patients, and upregulation of IL-6, IL-8, and matrix metallopeptidase 9 in the cortex and the cerebellum [62].

TNF-α, TNF receptor 1, IL-6, IL-1β, IL-8, and VEGF levels were significantly elevated, while IL-2, IL-4, IL-5, IL-10, IL-17, and IFN-γ were unchanged in the peripheral blood of ALS patients [63]. Granulocyte colony-stimulating factor, IL-2, IL-15, IL-17, monocyte chemotactic protein-1, macrophage inflammatory protein (MIP)-1α, TNF-α, and VEGF levels were significantly increased, but granulocyte-macrophage colony-stimulating factor, IFN-γ, IL-4, IL-5, IL-6, IL-7, IL-8, IL-10, IL-12p70, IL-13, MIP-1β, and regulated upon activation, normal T cell expressed, and presumably secreted (RANTES) levels were not significantly different in the CSF of ALS patients [64]. These results confirm the presence of inflammatory response in ALS. The levels of IL-1β, IL-8, IL-17, and monocyte chemoattractant protein were significantly increased in CSF of patients with CJD. The levels of IL-4, IL-10, and IL-1 receptor antagonist were significantly increased, while the level of TGF-beta 2 was decreased in CSF of patients with CJD [65,66,67,68].

The plasma concentrations of 13 cytokines were measured by multiplexed bead array immunoassay to investigate the association for verbal memory performance and HIV-seropositive individuals. The levels of IL-8 and IFN-γ were positively associated and the levels of IL-10 and IL-18 and hepatitis C infection were negatively associated with memory performance [69]. The levels of IL-8, monocyte chemotactic protein, granulocyte colony-stimulating factor, and induced protein-10 are elevated in the CSF of patients with neurocognitive impairment in HIV infection [70]. The deep white-matter lesions, driven by TNF-α, are associated with cognitive alteration, which is an indirect effect of HIV infection in the brain contributing to the development of HAND [71]. 

An increase in production of pro-inflammatory cytokines and a decrease in production of anti-inflammatory cytokines are correlated with a worse clinical outcome of stroke. TNF-α, IL-1β, IL-6, and IL-10 are inflammatory cytokines related to ischemic stroke [72]. CSF IL-6 is elevated within 24 hours in stroke patients and elevated in serum and plasma during the first week after stroke [73,74]. Lower levels of IL-10 were shown in plasma within 12 hours of stroke, and IL-10 levels are increased 24 hours after tissue-type plasminogen activator treatment [74,75]. IL-10 levels are decreased within 24 hours of stroke and increased over 72 hours post-stroke [76]. Pro-inflammatory cytokines were invariably increased in all major NDs, and anti-inflammatory cytokines were increased in AD, PD, HD, CJD, and HAND with low risk of bias for AD, PD, and CJD. Anti-inflammatory cytokines decreased in MS and SND and were unchanged in ALS, but presented a high risk of bias for MS, HD, HAND, and SND (Table 1).

## 4. Neurodegeneration-Induced Neuroinflammation, Chronic Inflammation, and Allostatic Loads

Neurodegeneration, microglia activation, and subsequent chronic neuroinflammation are central players in the pathogenesis of NDs. Activated microglial cells were observed to take an active part in Aβ plaque formation and growth. The microglial cells surround Aβ plaques to take up Aβ. Unresolved Aβ develops into clusters. Dying microglial cells release the accumulated Aβ clusters extracellularly, which contribute Aβ plaque growth [77]. Activated microglial cells were found to surround neurofibrillary tangle-bearing neurons, damaging dendrites and axons. Phagocytosed tau can be released in exosomes to promote tau propagation in adjacent cells [78]. Tau aggregates like prions and the aggregated tau activates the nod-like receptors P3-apoptosis-associated speck-like protein containing caspase activation and recruitment domains (CARD) (ASC) inflammasome, one of the sensors important for activation of innate immune response [79]. Early microglial activation and central and peripheral inflammation were detected in patients with dementia with LBs [80]. Furthermore, an animal study showed that injection of pre-formed α-syn fibrils triggered reactive microgliosis prior to nigral degeneration [81,82]. The level of cytoplasmic TDP-43 increases with age, and it was suggested that TDP-43 is a modulator of inflammation, which activates the nuclear factor kappa-light-chain-enhancer of activated B cells pathways, leading to a deregulation of innate immune response [83].

The neuroglial cells and peripheral macrophages are main contributors to neuroinflammation in HIV. Microglial cells and astrocytes are not only cellular reservoirs of HIV-1, but they are also activated by pro-inflammatory cytokines produced by HIV-infected peripheral monocytes and T cells [84]. Neurodegeneration is associated with chronic OS which harms genetic, lipid, and protein components to induce neuroinflammation. Microglial cells secrete harmful ROS and RNS, pro-inflammatory TNF-α, and excitotoxic glutamate due to the stimulus from Toll-like receptors through the aggregated proteins as in the case of AD, MS, PD, and ALS [85] (Figure 1d).

Chronic inflammation is characterized by the persistent activation of microglial cells that sustain the release of inflammatory cytokines, an increase of ROS and RNS, and neurotoxicity, all of which work together to perpetuate the inflammatory cycle, further prolonging inflammation, which is a detrimental pathogenesis for NDs [86]. Likewise, neurotropic viruses trigger long-term neuroimmune activation to underlying mechanisms of viral NDs [87].

Exaggerated stressful stimuli activate the sympathetic nervous system, leading to reduced sensitivity of GCs, thus amplifying the immune response in a viscous cycle. Proper responses of the organism to harmful stimuli can successfully lead to damage repair and subsequent recovery by proliferative inactivation and cellular elimination [88]. However, chronic immune response can be harmful and maladaptive, resulting in being unable to return to the healthy condition. The maladaptive phase of chronic stress response may refer to the exhaustion stage of Selye’s general adaptation syndrome or allostasis [89,90]. 

Harmful consequences are not due to exhaustion of the immune mechanism, but the stress mediators themselves damage the host, especially in the exhaustion stage. Sterling and Eyer proposed a concept of allostasis, which refers to the process of maintaining stability through change [91]. The allostatic systems which feature the adaptational cost of chronic exposure to inappropriate response to body, termed the allostatic load, eventually fail to perform normally, thus leading to diseases and eventually to death [92].

## 5. The Etiological Links behind Neurodegenerative Diseases: Tryptophan and Bioactive Kynurenines

Disturbance of tryptophan metabolism was reported in NDs [93]. Tryptophan is an essential amino acid which is used for protein synthesis and is a precursor to biosynthetic compounds, such as serotonin, melatonin, and nicotinamide adenine dinucleotide (NAD^+^), among others. A majority of research focused on areas of the serotonin pathway, but increasing attention is paid to the KYN pathway, which produces various bioactive compounds associated with inflammation, the immune system, the nervous system, and psychiatric diseases [94]. More than 95% of tryptophan is metabolized in the KYN pathway except for protein synthesis. Tryptophan (TRP) is converted to KYN by the hepatic rate-limiting tryptophan 2,3-dioxygenase (TDO) and ubiquitous rate-limiting indoleamine 2, 3-oxygenase (IDO) 1, which are induced by cortisol, and IFN-α, IFN-γ, and TNF-α, respectively. KYN is converted to anthranilic acid (AA) by the kynureninase, 3-hydroxy-l-kynurenine (3-HK) by the KYN-3-monooxygenase (KMO), and kynurenic acid (KYNA) by KYN aminotransferases (KATs). KATs also convert 3-HK to xanthurenic acid (XA). KYNA is an antagonist at the NMDA receptor. XA is converted to cinnabarinic acid (CA) by autoxidation. AA and 3-HK are converted to 3-hydroxyanthranillic acid (3-HAA) and further to picolinic acid (PA) and quinolinic acid (QA). 3-HK and QA are agonists at the NMDA receptor. QA is converted to NAD^+^, which is a feedback inhibitor of TDO [95] (Figure 2).

### 5.1. Tryptophan

Aberrant levels of TRP and disturbance of TRP metabolism are observed in patients suffering from NDs. Low circulating TRP levels are associated in elderly patients with NDs such as AD, PD, and HD [96]. TRP supplement was reported to enhance cognitive functions. Short-term TRP supplements improved serial attentions and reaction times and abstract visual memory, while chronic supplementation increased facial recognition memory and decreased baseline startle responsivity [97]. A TRP-fortified diet enhanced memory processes without improving the mood states in patients with MS [98]. 

### 5.2. Bioactive Kynurenines

The KYN pathway produces several small receptor agonists and bioactive molecules with a broad range of activities including neurotoxic, neuroprotective, anti-inflammatory, oxidative, antioxidative, and immune properties.

#### 5.2.1. Neurotoxic Kynurenines

QA is an excitotoxic NMDA receptor agonist which elicits hippocampal lesions like that of HD. Injections of QA into the striatum of rats induce selective degeneration of intrinsic striatal neurons with striatal atrophy and the lateral ventricle expansion, sparing intrinsic glial cells and myelinated axons of the internal capsule [94].

#### 5.2.2. Neuromodulatory Kynurenines

KYNA is a receptor antagonist of ionotropic α-amino-3-hydroxy-5-methyl-4-isoxazolepropionic acid (AMPA), kainite, and NMDA receptors [99]. Depending on the dose, however, KYNA exerts different actions at AMPA receptors. The micromolar concentrations are inhibitory, while the nanomolar concentrations are facilitatory by allosteric modulation of the AMPA receptor [100,101]. The actions of KYNA at the α-7 nicotinic acetylcholine receptor are controversial [102]. KYNA reduces glutamate release by binding to the G-protein-coupled receptor 35 (GPR35) in the microglial cells, macrophages, and monocytes to [94].

PA was shown to be neuroprotective. PA protects against QA- and kainic acid-induced neurotoxicity in the brain [103]. However, PA blocks the neurotoxic effects, but not the excitatory effects of QA. The mechanism of its anti-neurotoxic action is unclear but may be involved in zinc chelation or inhibition of nitric oxide synthase [104].

#### 5.2.3. Anti-Inflammatory Kynurenines

KYNA binds to the GPR35 expressed in the glia, macrophages, and monocytes to reduce pro-inflammatory cytokine release in cell lines [105]. AA and its 5-hydroxylated metabolites may possess potential anti-inflammatory properties. AA is metabolized to 3-HAA by a microsomal hydroxylase in mammalian liver. The anti-inflammatory reaction of AA and its related metabolites are associated with the fact that AA is a precursor of some nonsteroidal anti-inflammatory drugs such as mefenamic acid and diclofenac [106]. AA and 3-HAA were found to suppress pro-inflammatory cytokine IFN-γ, T- and B-lymphocyte cell proliferation, Th1 cell activity, and neurotoxicity induced by IL-1 or IFN-γ. They also invoke anti-inflammatory cytokine IL-10 [107]. PA can influence the immune response and has antifungal, antitumoral, and antibacterial activities [108].

#### 5.2.4. Reactive Oxygen Species

3-HK, 3-HAA, and QA are neurotoxic. 3-HK and 3-HAA generate ROS and might play a role in the regulation of OS [109]. An elevation of 3-HK levels was related to excitotoxic injury and is observed in patients with NDs [110]. The neurotoxic effect of the intermediates 3-HK and 3-HAA involves the generation of superoxide anion and hydrogen peroxide, which contribute to the oxidative processes implicated in the pathophysiology of meningitis [111]. 3-HK is the only KYN metabolite that is increased in plasma from vitamin B_6_-deficient subjects, suggesting that pyridoxal phosphate (PLP)-dependent enzymes are involved in its clearance [112]. The production of 3-HK and other TRP metabolites which filter ultraviolet light contribute to a gradual increase in yellowish pigmentation in the lens with age [113].

QA is also a free-radical metabolite. The pro-inflammatory cytokine IFN-γ activates IDO, formamidase, and KMO activities in human microglial cells and macrophages, resulting in increased QA synthesis. QA concentrations in the brain were reduced by anti-inflammatory steroid agent dexamethasone after immune stimulation [114].

#### 5.2.5. Antioxidant

KYNA is an antioxidant metabolite that scavenges ROS observed in various in vitro models and suppresses overshooting inflammation preventing tissue damage. KYNA can reduce FeSO_4_-triggered ROS toxicity primarily involving superoxide and hydrogen peroxide production [109,115]. Insufficient KYNA production may contribute to tissue damage and cell proliferation in inflammatory response in NDs.

#### 5.2.6. Immune Kynurenines

KYN, KYNA, xanthurenic acid, and cinnabarinic acid (CA) bind to the cytosolic aryl hydrocarbon receptor (AhR) transcription factor to suppress an adaptive immune response [116]. AhR activation is associated with TGF-β production, as well as IDO expression in dendritic cells, and it promotes differentiation of naive cluster of differentiation (CD) 4^+^ T-cells into immunosuppressive FoxP3^+^ regulatory T cells (Tregs) but not pro-inflammatory T helper (Th) 7 cells, through the AhR–Src–IDO1 pathway [117]. Activation of the KYN pathway suppresses effector T cells and induces Tregs, leading immune status to a tolerogenic state [118]. Increased KYN mediates inhibition of IL-2 signaling to reduce CD4^+^ T-cell survival [119]. IDO-expressing cells promote the differentiation of CD4^+^ T cells into Tregs expressing cytotoxic T-lymphocyte-associated protein 4, which downregulates immune responses [120]. Furthermore, higher KYNA production and lower KMO expression lead to dysfunctional effector CD4^+^ T-cell response [119]. NAD^+^ protects differentiated Th1, Th2, and Th17 from CD4^+^ T-cells, induces apoptosis of naïve CD4^+^ T-cells, and reduces the number of Tregs, but protects induced Tregs against apoptosis [121]. Thus, the KYN metabolic pathway enzymes and metabolites facilitate a shift toward tolerogenic T-cell functions.

PA activates macrophages. PA potently induces production of macrophage inflammatory protein-1α and -1β, which contributes to tumor suppression [122]. Interacting synergistically with IFN-γ, PA increases inducible nitric oxide synthase messenger RNA (mRNA) expression in macrophages leading to potent cytotoxic and cytostatic actions. PA-treated macrophages inhibit tumor growth and increase survival in cancer animal models. PA is an endogenous metal chelator for iron which decreased proliferation rates of tumor cells in vitro and in vivo but did not affect normal human cells at the same dosage [123].

### 5.3. Kynurenine Pathway Enzyme Activities

Inflammation activates several key enzymes in the KYN pathway. TDO in the liver, IDO 1 in the brain and peripheral tissues, and IDO 2 in the liver, kidney, and antigen-presenting cells (APCs) are the first rate-limiting enzymes of TRP metabolism [93]. The GC stress hormone, cortisol, activates TDO. The pro-inflammatory cytokines, IFN-α, IL-1β, IFN-γ, and TNF-α, activate IDO 1, while the anti-inflammatory cytokines, IL-2, IL-4, IL-10, and TGF-β, through IFN-γ, inhibit IDO 1 [93]. IDO 2 has a pro-inflammatory role, contributing to autoantibody production [95]. In addition, IDO-activated cells can alter the cytokine production of APCs to anti-inflammatory cytokines, TGF-β and IL−10, from pro-inflammatory cytokine, IL-12 [117].

The pro-inflammatory cytokine IFN-γ stimulates formamidase in human microglial cells and macrophages, leading to increased KYN synthesis. The activity of KATs is implicated in neurological and cognitive symptoms and the elderly [124,125]. A higher local KYN concentration is necessary for higher activity of KATs due to its low affinity. A cofactor, PLP, the active form of vitamin B_6_, and a cosubstrate, α-ketoacid, are required for KATs. [126]. A main source of PLP is food and degraded PLP-dependent enzymes by salvage pathway enzymes in humans. Genetic dysfunction of the salvage pathway enzymes and drug interactions of PLP or pyridoxal kinase result in convulsions and epileptic encephalopathy. A lower level of PLP is associated with neurological disorders including AD, PD, and epilepsy [127,128]. About 20% of the elderly are observed to have lower dietary vitamin B_6_ intake, and vitamin B_6_ supplementation improves cognitive performance in the elderly. It was proposed that folate, vitamin B_6_, and vitamin B_12_ are associated with cognitive performance [124,125]. 

The pro-inflammatory cytokine IFN-γ stimulates KMO activities in human microglial cells and macrophages, leading to increased QA synthesis. The activation of macrophages and glial cells induces the increased production of QA [129]. A lower KMO expression, in addition to higher KYNA production, is associated with dysfunctional effector CD4^+^ T-cell response, leading to suppression of adaptive immune response [130].

Microglial cells, which lack KATs but contain all enzymes that are involved in the successive conversion of KYN to QA, are believed to account for the local synthesis of 3-HK, 3-HAA, and QA. Moreover, microglia cells are responsible for the substantial upregulation of this major KP branch that is observed when the immune system is stimulated. KYNA synthesis, on the other hand, appears to occur almost exclusively in astrocytes, which lack KMO [131] (Figure 2).

### 5.4. Systematic Reviews on Kynurenines in Major Neurodegenerative Diseases

A systematic review on KYNs in dementia (major neurocognitive disorders) was reported previously [95]. Complementing the study, the systematic review was conducted on the levels of KYNs in MS, ALS, HAND, and CJD. Selection criteria and the risk of bias assessment are described in Appendix A. A total of 157 articles were found in our database search. One systematic review and 22 articles were evaluated for eligibility. Finally, six articles were included in this systematic review (Figure A2). The methodological quality and risk of bias assessment are presented in Table 1. Evidence levels of neurotoxic and neuromodulatory KYN levels were evaluated at high risk of bias for MS, ALS, and HAND, while no study was found in CJD (Table A2).

Errant levels of KYN metabolites were observed in patients with MS, ALS, CJD, and HAND. The KYN/TRP ratio was significantly increased in serum of MS patients. The QA levels were elevated, while NADH was decreased. 3-HK was found to be significantly higher in MS groups. The QA/KYNA ratio was higher in primary progressive MS, secondary progressive MS, and relapsing-remitting MS. KYNA levels were the highest in primary progressive MS, but lower in progressive MS [132]. Significantly elevated QA/KYN and QA/KYNA ratios were observed in the CSF of the relapsing subgroup of relapsing-remitting MS. TRP, KYNA, and QA levels were increased in primary progressive MS, while TRP and KYNA levels were decreased in secondary progressive MS [133]. KAT I and KAT II activities were significantly increased in the red blood cells (RBCs), and KYNA levels were significantly increased in the plasma of MS patients [134].

Significantly increased TRP, KYN, and QA in serum and CSF, and significantly decreased PA in serum were observed in ALS. Immunohistochemistry showed a significant increase in activated microglial cells expressing human leukocyte antigen DR isotype (HLA-DR) and increased neuronal and microglial expression of IDO and QA in the ALS motor cortex and spinal cord [135]. KYNA was significantly higher in CSF of the bulbar onset of ALS than that of healthy controls and the limb onset of the disease. KYNA was also higher in CSF of ALS patients with severe clinical status than controls. However, KYNA was significantly lower in serum of ALS patients with severe clinical status than healthy controls and ALS patients with mild clinical status. No significant difference was found between the whole ALS group of patients and healthy controls [136]. Differences in KYNA levels of the ALS subgroups may suggest a neuroprotective role of KYNA in disease progression and different pathogenesis in ALS. Regarding KYN levels in CJD, no article was found in the database. Postmortem human brain tissues from patients infected with HIV-1, identified with neuropsychiatric symptoms and histological evidence, were associated with elevated l-KYN and KYNA, and KAT I was prominently increased with respect to KAT II in both the frontal cortex and the cerebellum [137]. 

Neurotoxic KYNs were invariably increased in all major NDs, and neuromodulatory KYNs were decreased in MS, ALS, and HAND with high risks of bias. Neuromodulatory KYNs were increased in HAND with high risk of bias and mixed in MS and ALS depending on the subtypes (Table 2).

### 5.5. Synthesis of Inflammatory Cytokines and Bioactive Kynurenines: Signs and Symptoms

Of the major NDs, motor dysfunctions are significant in PD, MS, HD, ALS, HAND, and SND, while autonomic dysfunctions are significant in MS, and relatively significant in PD, HD, ALS, CJD, HAND, and SND. Psychobehavioral symptoms are significant in AD, MS, CJD, and HAND and relatively significant in PD, HD, ALS, and SND. All NDs presented evidence of innate inflammatory activation by increased pro-inflammatory cytokines. AD, PD, HD, CJK, and HAND showed activation of the secondary adaptive immune response by increased anti-inflammatory cytokines. However, MS and SND showed reduced levels of anti-inflammatory cytokines. Regardless of the activation or inactivation of the secondary adaptive immune response, the inflammatory profiles of major NDs described in this review deviated from a healthy state. Either causative or resultant of acute and chronic inflammation, the altered balance of bioactive KYNs was observed. Neurotoxic KYNs are increased in all major NDs. Neuromodulatory KYNs were increased in HAND and SND, decreased in AD, PD, and HD, and remained mixed in MS and ALS (Figure 3). 

## 6. Variations on the Theme? Commonalities in Symptomatology and Histopathology

The term ND was introduced relatively recently in 1965 and systemically defined in 2002 as age-related, incurable, and largely untreatable chronic progressive diseases of the CNS, including an ill-defined group of genetic and idiopathic disorders. However, the use of the large bucket term was challenged, and its use was proposed only for neurological diseases incurable and progressive. The term is not implied for a linkage to age-related, not necessarily inherited, disease, and it includes rapidly progressive diseases; therefore, it was proposed that “neuronal disease” may be a proper classifying term [138].

Neurological diseases are largely defined by their symptoms which are believed to reflect the anatomical location of their underlying pathology. NDs are a group of medical conditions nosologically classified according to primary clinical signs and symptoms, anatomical distribution of neurodegeneration, or histopathological and molecular findings [6]. Common features include a late-onset disease, neural atrophy, synaptic dysfunction, mitochondrial dysfunction, OS, neural cell apoptosis, and impaired protein homeostasis [139].

The common pathological course of the NDs is abnormal protein accumulation in the brain leading to topologically selective neurodegeneration, which characterizes signs and symptoms of diseases [6]. Nevertheless, some diseases have substantial overlaps in pathological and clinical presentations. In total, 30% of patients with AD and a higher percentage of patients with AD with LBs develop parkinsonism, whereas 30%–40% of PD patients presents dementia during their disease course. [140,141] (Figure 4a). Abnormal deposits of α-syn are pathognomonic features of LB disorders including PD, PD with dementia, and dementia with LBs (DLB) [142]. DLB and PD with dementia share AD-type pathology, and about 50% of PD with dementia is considered progression of the underling comorbidity with AD [143,144] (Figure 4b). The presence of LBs in the nucleus basilaris was declared pathognomonic to PD; however, they were detected in 100% of patients diagnosed with idiopathic PD by ubiquitin immunohistochemistry [145].

TDP-43 proteinopathy, pathognomonic to ALS, is detected in up to 50% of AD patients (Figure 4c) [146]. The plaques display different patterns and development with variable clinical course, but correlations with MS variants or clinical stages are unclear [147]. The formation of multiple plaques in the gray and white matter of the brain and spinal cord is a hallmark in MS; however, it was reported that neurodegeneration can occur independent of white-matter demyelination in a subtype called myelocortical MS [148].

The bilateral symmetrical atrophy with flattening of convex contour is unique to HD, but the atrophic change is also found in HAND. The ballooned nerve cells abundant in HD are found in tauopathy such as in AD, but the distribution is distinct from that of HD [149]. Aβ deposition is also observed in about 10% of CJD, a rare transmissible prion disease starting with AD-like cognitive symptoms, but rapidly progressing to dementia, coma, and death in a year [150] (Figure 4c). Progressive and non-random spreading of Aβ deposits led to a hypothesis that Aβ becomes a prion in the pathogenesis of AD [151]. Tau is also considered to function as a prion [152].

Diffuse intracellular Aβ plaques were noted in the temporal and frontal lobes and associated with age and HIV status. Increased hyperphosphorylated tau in the hippocampus was reported in HIV-infected patients on cART (Figure 4c). Thus, differential diagnosis of AD in patients with HAND is a challenging issue [153].

SND shares many similarities to AD pathology such as abnormal Aβ deposition, neural loss, and glial disturbance observed in animal and human studies. SND is linked to the pathogenesis of AD and PD in sequelae of stroke and, thus, it was proposed that stroke triggers neurodegeneration [154] (Figure 4c).

The commonalities were also found on a molecular level and studied in silico [139]. A family history of the disease can be detected in HD, but in only 1%–10% of AD, PD, and ALS cases. The common disease–common variant hypothesis proposes that the genes of common variants such as single nuclear polymorphisms, coding regulatory sequences, lead to susceptibility of complex polygenic diseases [155]. OS damages RNA to cause dysregulation of gene expression affecting inflammatory processes, the ubiquitin–proteasome system, energy production, cell-cycle regulation, and glutamate excitotoxicity [156]. Many proteins play an important part in the regulation of NDs. Genetic overlap was reported for risk factors and onset modifiers. The apolipoprotein E gene, located on chromosome 19 and coding a 299-amino-acid glycoprotein, is considered to influence the progression of AD, PD, MS, SND, and type II diabetes mellitus, supporting a hypothesis of shared mechanisms in the common diseases [157].

## 7. Conclusions

Scientists and physicians tend to reduce the cause of a disease into the result of a single unique starting point, and much attention is paid to striking findings, signs, and symptoms developed in the course of disease, even if there may be no link to causality. These attitudes often lead to errors [158]. NDs may be illnesses with a dynamic state in which pathophysiological alternation and clinical symptoms develop in a certain range over time. Many chronic diseases have a multifactorial origin associated with factors of genetic, environmental, socioeconomic, cultural, and personal lifestyle predispositions [159,160,161]. The links between genetic and environmental factors were documented by mechanisms of epigenetics such as DNA methylation, DNA hydromethylation, chromatin remodeling, histone modifications, non-coding RNAs, and microRNAs, among others. Aging and CNS diseases are considered more susceptible to the disturbance of epigenetics [162]. Both endogenous and exogenous factors can stay under mutual influence and suitable control in a range, and they remain innocent up to a certain amount and duration to maintain homeostasis [89]. However, a prolonged inappropriate body response at the expense of adaptational cost leads to physiological malfunction and eventually to disease [91]. A multifactorial disease may break out and progress in a certain path via a group of factors not common among patients but which present similar pathognomonic signs and symptoms [163]. Thus, the attempt to discover a unique mechanistic cause for a disease is considered ill-founded and misleading.

In the pathogenesis of multifactorial diseases, the cause–effect relationships can become bidirectional, that is, a cause can become an effect and vice versa in the course of disease development [164]. Constantly changing numerous factors may form an effective causal complex which plays an initiation role in multiple reciprocal causations, together leading to a disease-prone state [165]. A retrospective cohort study on a set of metabolomic biomarkers may help identify the chemical fingerprints of preclinical stage and become of great value for the discovery of new drugs to slow the progression to disease. A proximal cause consisting of numerous factors not necessarily common to patients evolves into an ultimate cause, which eventually leads to a full-blown ND [166]. Such complex etiological and progressive situations at least partly contribute to the heterogeneity of multifactorial NDs, and they make it difficult to analyze and leave a considerable amount of uncertainty, which may be a major attribute to the failure of numerous target-approached clinical trials and the tendency toward researches focusing on palliative measures (Figure 5a).

In search of a cause of multifactorial NDs, multi-causality and probabilistic determinism elaborated classical causal criteria which evidence a relationship between a presumed cause and an observed effect [167]. Analytical approximation further contributes to the better understanding of a role of medical statistics in a causal relationship [168]. Incorporation of a reference case may ease reaching an interpretation of meaningful comparisons by integrating standardization [169]. Furthermore, causality is not simply reducible to difference-making and mechanistic interpretations which just inform causal assessment. An inferential map drawn with elements of causal relationships enables inferences on explanations, managements, and predictions according to the epistemic causality search, which assesses evidence and builds a hierarchy of evidence in evidence-based medicine [170] (Figure 5b).

## Figures and Tables

**Figure 1 ijms-21-02431-f001:**
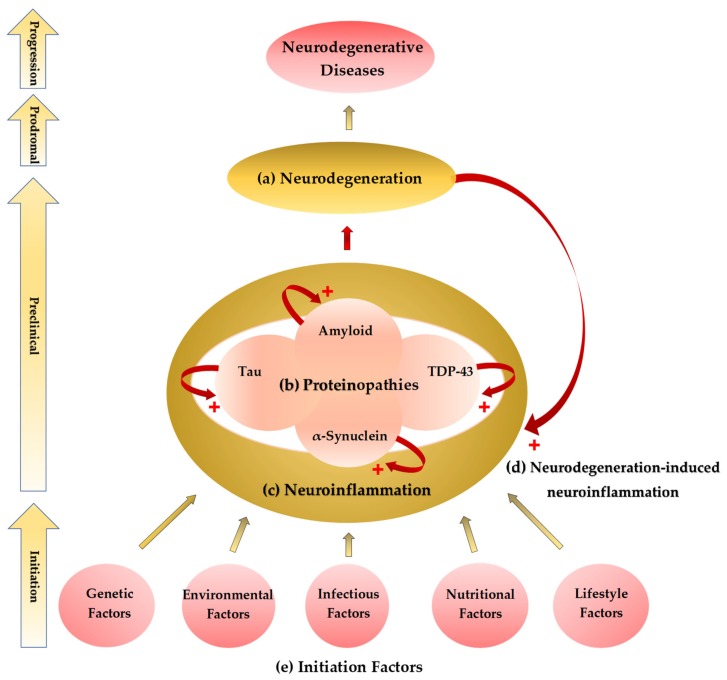
Initiation–progression hypothesis of neurodegenerative diseases. (**a**) A common pathological finding of neurodegenerative diseases (NDs) is neurodegeneration such as neural loss and gliosis. The location and distribution of neurodegeneration present corresponding neurological and psychiatric signs and symptoms. (**b**) Histopathological studies showed abnormal proteinaceous deposits in the degenerative neural tissues. Proteinopathies include amyloidosis, tauopathy, synucleopathy, and transactive response DNA-binding protein (TDP)-43 proteinopathy. (**c**) Activation of glial cells and astrocytes causes neuroinflammation, most probably the last common pathway leading to neurodegeneration. Neuroinflammation potentiates the accumulation of the abnormal proteinaceous materials in the brain tissue. (**d**) Neurodegeneration also causes neuroinflammation forming a cyclic potentiation. (**e**) The proteinopathy–neuroinflammation–neurodegeneration cycle can be triggered by multifactorial initiation factors not yet clearly defined. The genetic, environmental, infectious, nutritional, and lifestyle components are major factors which contribute to the initiation process of the NDs.

**Figure 2 ijms-21-02431-f002:**
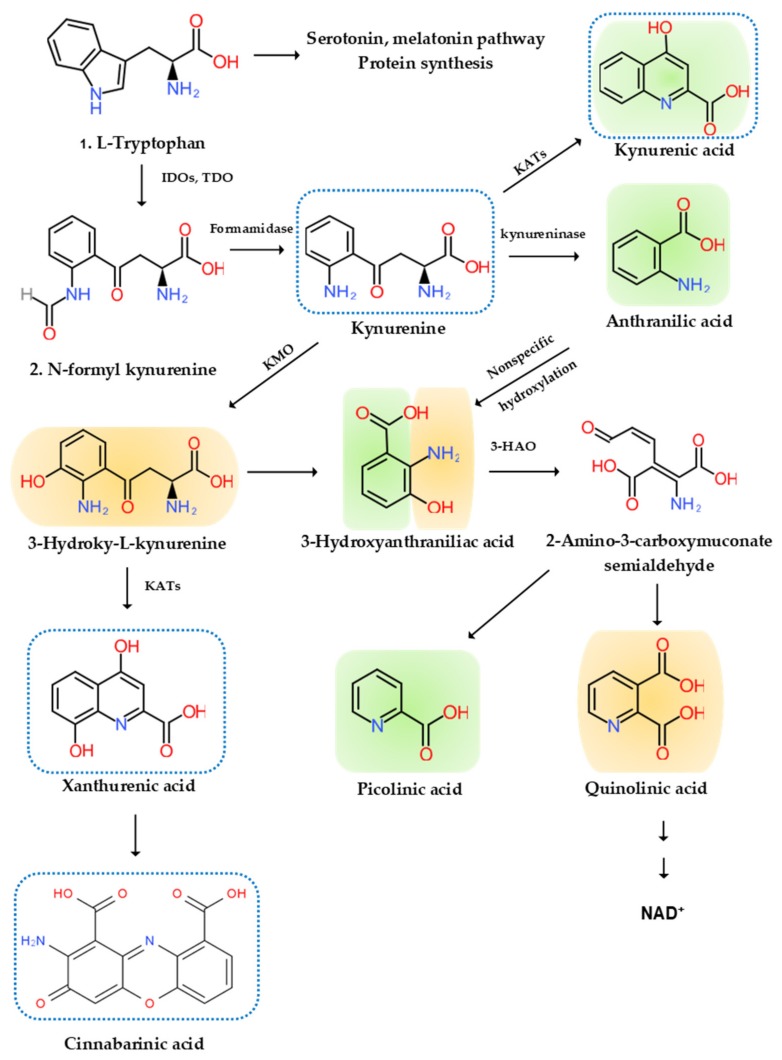
l-Tryptophan metabolism and the kynurenine pathway. The indole ring of l-tryptophan (TRP) is oxidized by the TRP dioxygenase (TDO) and the indolamine-2,3-dioxygenase (IDO) to produce *N*-formyl kynurenine (KYN). *N*-formyl KYN is converted by the formamidase to KYN, a substrate of three downstream metabolites: anthranilic acid (AA) by the kynureninase, 3-hydroxy-KYN (3-HK) by the KYN-3-monooxygenase (KMO), and kynurenic acid (KYNA) by KYN aminotransferases (KATs), which also convert 3-HK to xanthurenic acid (XA). XA is converted to cinnabarinic acid (CA) by autoxidation. AA and 3-HK are converted by 3-hydroxyanthranilate oxidase to 3-hydroxy-AA (3-HAA). 3-HAA is converted by 3-hydroxyanthranilate dioxygenase to 2-amino-3-carboxymuconate semialdehyde, which is further transformed into picolinic acid (PA) and quinolinic acid (QA). QA is further converted to nicotinic acid, nicotinic acid adenine dinucleotide, and nicotinamide dinucleotide (NAD^+^). Neurotoxic, oxidative KYNs are shown in orange color, neuromodulartory anti-inflammatory and antioxidant KYNs are shown in green color, and aryl hydrocarbon receptor agonists are represented by a blue dotted line.

**Figure 3 ijms-21-02431-f003:**
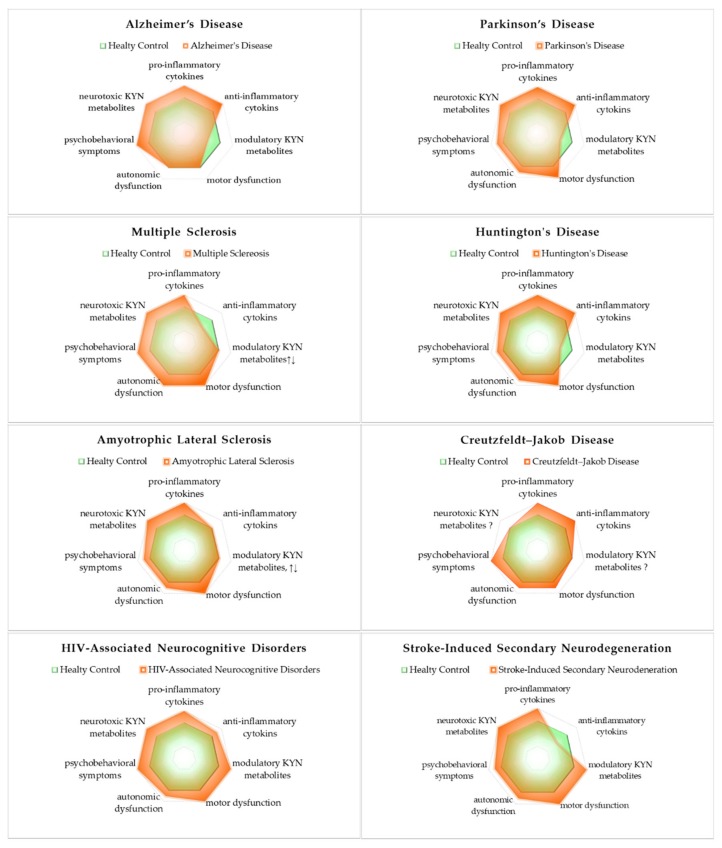
The status of inflammatory cytokines and kynurenines: signs and symptoms of neurodegenerative diseases. All neurodegenerative diseases significantly increased pro-inflammatory cytokines and neurotoxic cytokines. The involvement of three representative symptoms of neurological diseases is described in lower axes of the profiles: motor dysfunction, autonomic dysfunction, and psychobehavioral symptoms.

**Figure 4 ijms-21-02431-f004:**
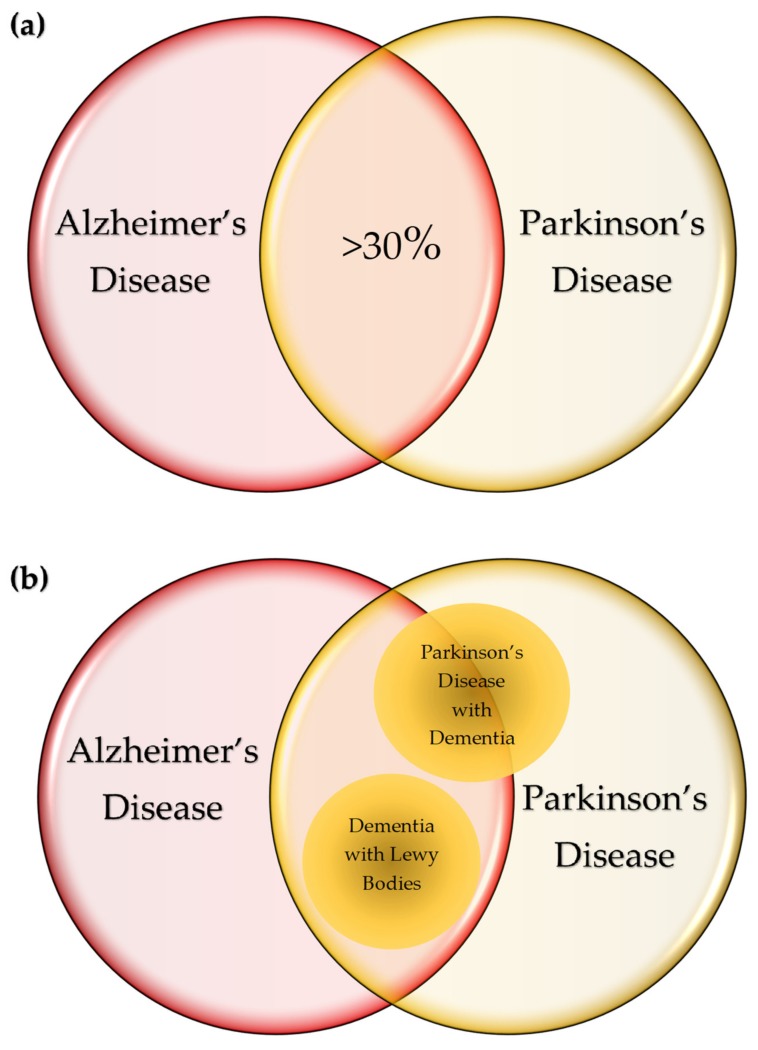
Commonalities of neurodegenerative diseases. (**a**) In total, 30% of patients with Alzheimer’s disease (AD) develop parkinsonism, whereas over 30% of patients with Parkinson’s disease (PD) present dementia during their disease course. (**b**) Lewy body (LB) disorders include PD, PD with dementia, and dementia with LBs (DLB). DLB and PD with dementia share AD-type pathology and about 50% of PD with dementia is considered progression of the underling comorbidity with AD. (**c**) The amyloid deposits in the neurons and glial cells are characteristic to AD, and tauopathy is most prevalent in AD. The tauopathy of AD is considered secondary to the deposition of amyloid-β (Aβ). Pathognomonic to amyotrophic lateral sclerosis (ALS), TDP-43 proteinopathy is detected in up to 50% of patients with AD. The ballooned nerve cells abundant in Huntington’s disease (HD) are found in tauopathy such as in AD. Amyloid deposition is also observed in about 10% of Creutzfeldt–Jakob disease (CJD), starting with AD-like cognitive symptoms, and rapidly progressing to dementia and death. Diffuse intracellular amyloid plaques are associated with age and HIV-associated neurocognitive disorders (HAND). Increased hyperphosphorylated tau was reported in HIV-infected patients on combination antiretroviral therapies (cART). Stroke-induced secondary neurodegeneration (SND) shares many similarities to AD such as abnormal amyloid deposition.

**Figure 5 ijms-21-02431-f005:**
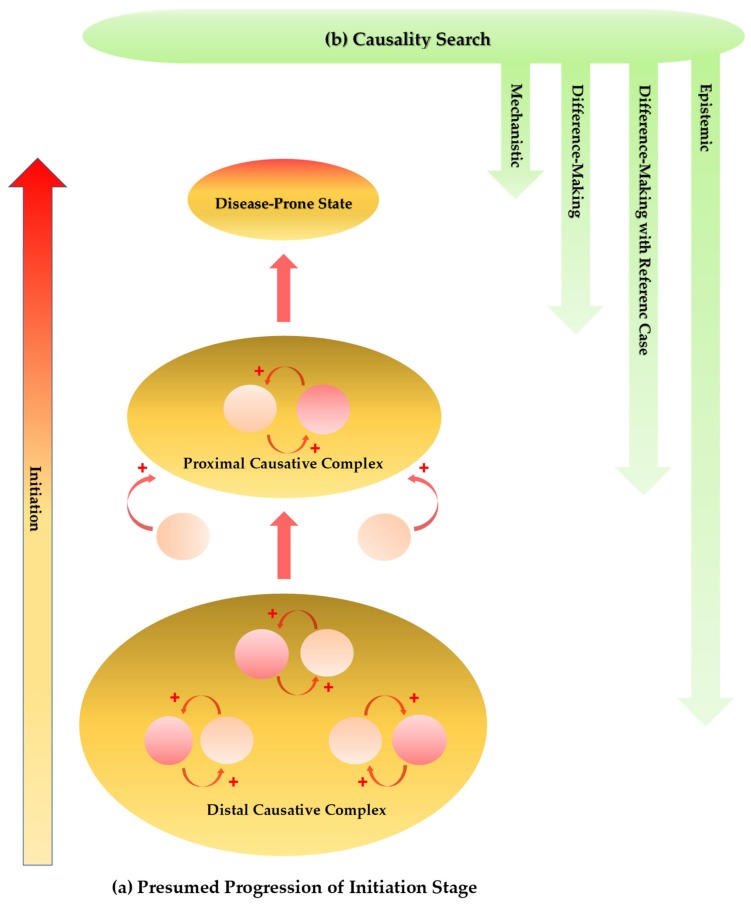
Presumed progression of initiation stage and approaches for causality search of multifactorial neurodegenerative diseases. (**a**) The genetic, environmental, infectious, nutritional, and lifestyle components are major factors in the initiation process. The cause–effect relationships can become bidirectional. A cause becomes an effect and an effect becomes a cause, playing an initiation role in multiple reciprocal causations. Constantly changing numerous factors form an effective causative complex. The distal causative complex influences and develops the proximal causative complex in progression of the initiation phase, leading ultimately to a disease-prone state. (**b**) The causal assessment is performed by multi-causality and probabilistic determinism establishing classical causal criteria. Mechanistic and difference-making interpretations inform causality assessment. Incorporation of a reference case may ease reaching an interpretation of meaningful comparisons. The epistemic causality search assesses evidence which cannot be reducible by mechanistic and difference-making interpretations.

**Table 1 ijms-21-02431-t001:** Systematic synthesis of pro-inflammatory and anti-inflammatory cytokine levels in neurodegenerative diseases. ↑: increase; ↓: decrease; -: unchanged. HIV—human immunodeficiency virus.

Diseases	Pro-Inflammatory Cytokines	Anti-Inflammatory Cytokines
Alzheimer’s disease	↑	↑
Parkinson’s disease	↑	↑
Multiple sclerosis	↑	↓
Huntington’s disease	↑	↑
Amyotrophic lateral sclerosis	↑	-
Creutzfeldt–Jakob disease	↑	↑
HIV-associated neurocognitive disorders	↑	↑
Stroke-induced secondary neurodegeneration	↑	↓

**Table 2 ijms-21-02431-t002:** Systematic synthesis of neurotoxic and neuromodulatory kynurenine levels in neurodegenerative diseases. ↑: increase; ↓: decrease; ↑ ↓: mixed result; ?: unknown.

Diseases	Neurotoxic Kynurenines	Neuromodulatory Kynurenines
Alzheimer’s disease	↑	↓
Parkinson’s disease	↑	↓
Multiple sclerosis	↑	↑ ↓
Huntington’s disease	↑	↓
Amyotrophic lateral sclerosis	↑	↑ ↓
Creutzfeldt–Jakob disease	?	?
HIV-associated neurocognitive disorders	↑	↑
Stroke-induced secondary neurodegeneration	↑	↑

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
