# Peer review of "Exploring the Etiological Links behind Neurodegenerative Diseases: Inflammatory Cytokines and Bioactive Kynurenines"

_ijms, 2020, doi:10.3390/ijms21072431_

Round 1
Reviewer 1 Report
The manuscript titled "The Scene Behind Neurodegenerative Diseases: Inflammatory Cytokines and Bioactive Kynurenines” proposed to review the literature on inflammatory cytokines, tryptophan metabolites, and kynurenines (KYNs) in human samples and their link to various neurodegenerative diseases (NDs). The authors conducted a literature search using PubMed/MEDLINE and applying appropriate search terms and filters. The authors concluded that there is a strong link between inflammation, neurotoxic KYNs, and neurodegeneration, and highlighted the limitations of current diagnostic criteria for NDs and suggested the need for inclusion of additional preclinical biomarkers as initiation factors of NDs, such as cytokines and KYNs. Such a study could be very interesting for the field and could put together, for physicians and scientists from several fields, a concise review linking neurodegenerative processes to various inflammatory markers (focusing on cytokines, KYNs, and others). Even though the concept of the review is very interesting, the organization of the topics/subtopics and the writing of the manuscript were poorly executed. Bellow are a few suggestions and examples supporting these comments:
- The authors start the manuscript with a review of NDs (Alzheimer’s disease, Parkinson’s disease, amyloid lateral sclerosis, Huntington disease, and others). This review is not the focus of the manuscript and was poorly written. The figure that tries to link the Initiation Factors to Neuroinflammation, Proteinopathies, and Neurodegeneration (Figure 1) is confusing. For example, the letters that would suggest the order the reader should follow to understand the figure (a<b<c<d<e) seems to be out of order and it is difficult to determine the flow of information.
- Whereas a review of NDs seemed unnecessary (as it is currently written), a review of bioactive kynurenines and a proper introduction to the topic was not presented to the reader. The authors “jumped” into the topic of “aberrant levels of tryptophan …” without laying the ground for the reader to what KYNs are in the first place and why they are important.
- A few statements along the text need to be reviewed. Below are two examples: In line 184 the authors stated: “Microglia are the CNS-infiltrating macrophages …” seeming to suggest that all microglia are peripheric and infiltrated the blood brain barrier. Additionally, in lines 34/35 the authors stated: “Neurodegenerative disease (ND) is a range of progressive and incurable conditions which affect the neurons in the brain and/or spinal cord …” seeming to suggest that all NDs are incurable/untreatable and affect the neurons only.
There are innumerous errors along the text (typos, grammar mistakes, punctuation errors, etc.) that make the text difficult to follow. A thorough review is needed.
Author Response
Thank you for the precious comments and valuable suggestions. We made some adjustment and corrections as follows:
- Thank you for the valuable suggestions. The abstract was revised to clarify the purpose of this review, present the current issues and suggest possible approaches. Brief reviews on pathological and clinical aspects were included in the review to present commonalities in neurodegenerative diseases according to current pathological and diagnostic criteria, which are less useful for disease prevention and drug development. We also intended to make it informative and readable to molecular biologists who focus on molecules as well as physicians who diagnose and treat patients. The inferential method was presented to explore a role of inflammation and involvement of kynurenine metabolism among many other potential factors. Figure 1 was edited accordingly, to be able to follow a logical order and directions downward and rightward.
- Thank you for the precious comments. An introductory phrase was added in the beginning of section, the kynurenine pathway and its metabolites were described in detail.
- Thank you for the precious comments. The sentences were corrected accordingly.
Thank you for the precious comments. We corrected it at our best.
Reviewer 2 Report
In their review titled "The Scene behind Neurodegenerative Disease: Inflammatory Cytokines and Bioactive Kynurenines" Masura et al. show the role of the inflammatory cytokines and kynurenines in the neurodegenerative process, including the main neurodegenerative diseases such as PD, AD and HD, among others.
The review is well written and important research topic. Just minor concerns are as follows:
You have to improve the abstract according to your review, there is a lot of lack of information. Furthermore, you have to improve English grammar, acronyms, and so on.
Consider changing "etc" for "among others".
Eliminate a from “a further” line 59.
Changes "injury" for "injuries" line 60.
Acronym of Lewy bodies (LB).
"Hippocampus" for "the hippocampus".
Revise the space in the "tangle -bearing neurons" 279.
Consider changing "associated to" for "associated with".
"have shifted away from healthy State" for "From a healthy State", Line 496.
"in silico" in cursive, Line 553.
Acronym for Oxidative stress like (OS).
-Abstract
Change the abstract information according to the introduction you have to include neurodegeneration term and neurodegenerative disease in common...
-Introduction
Concrete a little bit more which symptoms for each disease.
Consider adding the prevalence of PD, and for the rest of NDs
Define at least two times the term neurodegeneration in a different section. Eliminate one of them.
There's redundant information in each section, try to correct this.
Consider improving the quality of figure 3 I can not see the letters too much well.
The term epigenetic is epigenetics. Furthermore, try to add a little information about epigenetics before. Perhaps, when you talk about initiators and lifestyle just a few lines and add some references. For instance, Understanding epigenetics in Neurodegeneration (Griñán-Ferré et al., 2018). Due to you only talk in the conclusions and it is not enough.
Congratulation for the manuscript.
Author Response
Thank you for the precious comments and valuable suggestions. We made some adjustment and corrections as follows:
In their review titled "The Scene behind Neurodegenerative Disease: Inflammatory Cytokines and Bioactive Kynurenines" Masura et al. show the role of the inflammatory cytokines and kynurenines in the neurodegenerative process, including the main neurodegenerative diseases such as PD, AD and HD, among others.
The review is well written and important research topic. Just minor concerns are as follows:
You have to improve the abstract according to your review, there is a lot of lack of information. Furthermore, you have to improve English grammar, acronyms, and so on.
Thank you for the valuable comments. The abstract was revived to clarify the purpose of this review.
Consider changing "etc" for "among others".
Eliminate a from “a further” line 59.
Changes "injury" for "injuries" line 60.
Thank you for the precious comments. They were corrected accordingly.
Acronym of Lewy bodies (LB).
Thank you for the precious comments. The acronym was added accordingly.
"Hippocampus" for "the hippocampus".
Revise the space in the "tangle -bearing neurons" 279.
Consider changing "associated to" for "associated with".
"have shifted away from healthy State" for "From a healthy State", Line 496.
"in silico" in cursive, Line 553.
Thank you for the precious comments. They were corrected accordingly.
Acronym for Oxidative stress like (OS).
Thank you for the precious comments. The acronym was added accordingly.
-Abstract
Change the abstract information according to the introduction you have to include neurodegeneration term and neurodegenerative disease in common...
Thank you for the valuable suggestion. We define neurodegeneration as a progressive loss of structure or function of neurons, which is a pathological condition in a group of neurodegenerative diseases. We use the term neurodegenerative disease as a group of neurological diseases which are caused by a structural and functional loss of neurons, neurodegeneration. We noticed it causes some confusion such as stroke-induced secondary neurodegeneration (SND), which we use the term as a disease entity. We rephrased a couple of sentences in Abstract and defined the term neurodegeneration in Introduction, and clarified SND to avoid confusion.
-Introduction
Concrete a little bit more which symptoms for each disease.
Thank you for the thoughtful comments. Brief descriptions were added in section 2.
Consider adding the prevalence of PD, and for the rest of NDs
Thank you for the thoughtful comments. The prevalence of PD was added and the rest of NDs was described as a group of neurological diseases to present the second leading cause of deaths.
Define at least two times the term neurodegeneration in a different section. Eliminate one of them.
Thank you for the thoughtful comments. It was omitted accordingly.
There's redundant information in each section, try to correct this.
Thank you for the valuable suggestion. We tried to make concise.
Consider improving the quality of figure 3 I can not see the letters too much well.
Thank you for the thoughtful comments. We edited the figure with lager size of the font.
The term epigenetic is epigenetics. Furthermore, try to add a little information about epigenetics before. Perhaps, when you talk about initiators and lifestyle just a few lines and add some references. For instance, Understanding epigenetics in Neurodegeneration (Griñán-Ferré et al., 2018). Due to you only talk in the conclusions and it is not enough.
Thank you for the valuable suggestions and reference. It was corrected accordingly. The reference was added regarding exogenous factors and epigenetics.
Congratulation for the manuscript.
We gratefully appreciate your helpful comments, valuable suggestions and expert opinions. We sincerely hope that this revised manuscript could be considered for publication in International Journal of Molecular Sciences.
Reviewer 3 Report
Manuscript ID: ijms-745787
Title: The Scene Behind Neurodegenerative Diseases: Inflammatory Cytokines and Bioactive Kynurenines
This is an excellent review of neurodegenerative diseases from the point of inflammation, emphasizing especially the role of kynurenine. Since the contents are well-summarized, the manuscript is almost ready for the publication. Below are some suggestions, which may help to improve the value of this review.
(1) Any researches describing the molecular mechanisms explaining how various clinical symptoms are caused by seemingly common inflammatory abnormalities should be reviewed.
The section 2, "The Common Theme: Proteinaceous Deposits, Neurodegeneration, Neuropathies", line 72 is an excellent summary of the clinical features of many neurodegenerative diseases. The major concerns are the molecular mechanisms causing such various symptoms from seemingly common pathological changes including inflammatory cytokines. This point should be reviewed in addition to the summary of clinical features. In section 6 "Variations on the theme? Commonalities and heterogeneity", line 506, the discussion of heterogeneity should be expanded with an explanation illustration.
(2) The researches on kynurenines should be briefly summarized including its history.
Before section 5.1 "Tryptophan" line 316, a brief summary of kynurenines reseaches including its physiological functions (not limited to CNS, but also peripheral tissues) and historical points of the research is recommended.
(a) Title: The Scene Behind Neurodegenerative
"Scene" is not scientific. The more descriptive title is preferred. "The prevalent etiological procedure behind neurodegenerative diseases" and so on.
(b) line 14 "The proteinopathy-induced neurodegeneration leading to anatomically corresponding neuropathies, is a clinical vignette."
This sentence is difficult to understand. Especially "clinical vignette" cannot be understood. "is a major concern"? I recommend this sentence to be rewritten.
(c) line 60 "This review article overviews the pathological and clinical observation of NDs including neurodegeneration which is caused by histopathological insults and responsible for specific signs and symptoms, and neuroinflammation which is potentiated by neurodegeneration and exacerbating proteinopathies."
The above sentence "neuroinflammation is potentiated by neurodegeneration" means the cause of neuroinflammation is neurodegeneration. However in line 179 "Neuroinflammation: A Common Prelude to Neurodegeneration", it is described that neuroinflammation causes neurodegeneration. This seems to be contradiction.
(d) Line 447 "5.4. Synthetic Reviews on Kynurenines in Major Neurodegenerative Diseases"
"Synthetic" is not understandable here. It may be "systematic".
End of File
Author Response
Thank you for the precious comments and valuable suggestions. We made some adjustment and corrections as follows:
Manuscript ID: ijms-745787
Title: The Scene Behind Neurodegenerative Diseases: Inflammatory Cytokines and Bioactive Kynurenines
This is an excellent review of neurodegenerative diseases from the point of inflammation, emphasizing especially the role of kynurenine. Since the contents are well-summarized, the manuscript is almost ready for the publication. Below are some suggestions, which may help to improve the value of this review.
<Major Points>
(1) Any researches describing the molecular mechanisms explaining how various clinical symptoms are caused by seemingly common inflammatory abnormalities should be reviewed.
The section 2, "The Common Theme: Proteinaceous Deposits, Neurodegeneration, Neuropathies", line 72 is an excellent summary of the clinical features of many neurodegenerative diseases. The major concerns are the molecular mechanisms causing such various symptoms from seemingly common pathological changes including inflammatory cytokines. This point should be reviewed in addition to the summary of clinical features.
Response: We appreciate the valuable suggestions. Brief molecular mechanisms leading to neurodegeneration and neuropathies were added regarding amyloid-β, tau protein, alpha-synuclein, plaques, ballooning cell death, TDP-43 protein, and prion protein.
In section 6 "Variations on the theme? Commonalities and heterogeneity", line 506, the discussion of heterogeneity should be expanded with an explanation illustration.
Response: We appreciate the valuable suggestions. Three figures were added to explain the commonalities visually. Hopefully they help readers to understand the section. Heterogeneity was omitted from the section and included in Conclusion.
(2) The researches on kynurenines should be briefly summarized including its history.
Before section 5.1 "Tryptophan" line 316, a brief summary of kynurenines reseaches including its physiological functions (not limited to CNS, but also peripheral tissues) and historical points of the research is recommended.
Response: We appreciate the valuable suggestions. A brief introductory description was added regarding the tryptophan metabolism and the kynurenine pathway and its metabolites were summarized in the beginning of the section and the figure of the kynurenine pathway follows.
<Minor Points>
(a) Title: The Scene Behind Neurodegenerative
"Scene" is not scientific. The more descriptive title is preferred. "The prevalent etiological procedure behind neurodegenerative diseases" and so on.
Response: We appreciate the thoughtful comment. The title was changed to “Exploring the Etiological Links Behind…..”
(b) line 14 "The proteinopathy-induced neurodegeneration leading to anatomically corresponding neuropathies, is a clinical vignette."
This sentence is difficult to understand. Especially "clinical vignette" cannot be understood. "is a major concern"? I recommend this sentence to be rewritten.
We appreciate the thoughtful comment. ”clinical vignette” is omitted and the sentence was rephrased to “A common clinical course is the proteinopathy-induced neurodegeneration leading to anatomically corresponding neuropathies.”
(c) line 60 "This review article overviews the pathological and clinical observation of NDs including neurodegeneration which is caused by histopathological insults and responsible for specific signs and symptoms, and neuroinflammation which is potentiated by neurodegeneration and exacerbating proteinopathies."
The above sentence "neuroinflammation is potentiated by neurodegeneration" means the cause of neuroinflammation is neurodegeneration. However in line 179 "Neuroinflammation: A Common Prelude to Neurodegeneration", it is described that neuroinflammation causes neurodegeneration. This seems to be contradiction.
We appreciate the thoughtful comment. We aim to describe the pathological process in which neurodegeneration also contributes to neuroinflammation in a cyclic manner. To clear the confusion, Figure 1 was revised to follow logical sequence downward and rightward.
(d) Line 447 "5.4. Synthetic Reviews on Kynurenines in Major Neurodegenerative Diseases"
"Synthetic" is not understandable here. It may be "systematic".
We appreciate the thoughtful comment. It was corrected accordingly.
End of File
We gratefully appreciate your helpful comments, valuable suggestions and expert opinions. We sincerely hope that this revised manuscript could be considered for publication in International Journal of Molecular Sciences.
Round 2
Reviewer 1 Report
The revised version of the manuscript titled “Exploring the Etiological Links Behind Neurodegenerative Diseases: Inflammatory Cytokines and Bioactive Kynurenines” addressed the main concerns raised after the review of the original draft. The alterations have improved the organization and presentation of the various topics discussed in the manuscript as well as readability. The addition of a few statements explaining the link between the pathological hallmarks (and symptoms) of each neurodegenerative disease and activation of immune response and inflammation (i.e. lines 189-194 and lines 249-255) has made the “story” more coherent. I would like to point out, however, that some minor points still need to be addressed. First, in lines 124/125 the authors state that “The Aβ deposition in the interneurons leads to microglial activation …” and this is not correct. Aβ plaques are formed extracellularly and accumulation of Aβ is certainly not limited to interneurons. Also, small errors can still be found along the text and should be corrected. I have included three examples below (although there are others):
Lines 40/41: “More than 10 million people worldwide are suffered from PD and the incidence increases with age”
Lines 639/640: “Aberrant levels of TRP and disturbance of TRP metabolism have been observed in patients suffered from NDs.”
Line 1322 (Figure 4): “The tauopathy of AD is considered the secondary to the deposition of Aβ.”
Such minor mistakes, however, do not affect the overall contribution of this work and can be easily addressed.